# Category-specific perceptual learning of robust object recognition modelled using deep neural networks

Hojin Jang[1]*, Frank Tong[2]*

**1** Department of Brain and Cognitive Engineering, Korea University, Seoul, South Korea, **2** Department of Psychology and Vanderbilt Vision Research Center, Vanderbilt University, Nashville, Tennessee, United States of America

* hojin4671@korea.ac.kr (HJ); frank.tong@vanderbilt.edu (FT)

## Abstract

Object recognition in real-world environments requires dealing with considerable ambiguity, yet the human visual system is highly robust to noisy viewing conditions. Here, we investigated the role of perceptual learning in the acquisition of robustness in both humans and deep neural networks (DNNs). Specifically, we sought to determine whether perceptual training with object images in Gaussian noise, drawn from certain animate or inanimate categories, would lead to category-specific or category-general improvements in human robustness. Moreover, might DNNs provide viable models of human perceptual learning? Both before and after training, we evaluated the noise threshold required for accurate recognition using novel object images. Human observers were quite robust to noise before training, but showed additional category-specific improvement after training with only a few hundred noisy object examples. In comparison, standard DNNs initially lacked robustness, then showed both category-general and category-specific learning after training with the same noisy examples. We further evaluated DNN models that were pre-trained with moderately noisy images to match human pre-training accuracy. Notably, these models only showed category-specific improvement, matching the overall pattern of learning exhibited by human observers. A layer-wise analysis of DNN responses revealed that category-general learning effects emerged in the lower layers, whereas category-specific improvements emerged in the higher layers. Our findings provide support for the notion that robustness to noisy visual conditions arises through learning, humans likely acquire robustness from everyday encounters with real-world noise, and additional category-specific improvements exhibited by humans and DNNs involve learning at higher levels of visual representation.

**Data availability statement:** All data and codes employed in this study will be available on the Open Science Framework at https://osf.io/zvd7g/.

**Funding:** This research was supported by the following grants from the National Institutes of Health: R01EY035157 and R01CA240274 to FT, and P30EY008126 to the Vanderbilt Vision Research Center. Additional support was provided by the National Research Foundation of Korea, funded by the Korean government (RS-2024-00451866 to HJ). The funders had no role in study design, data collection and analysis, decision to publish, or preparation of the manuscript.

**Competing interests:** I have read the journal's policy and the authors of this manuscript have the following competing interests: It should be noted that the noise-training methods used in this study were described in a patent issued with the U.S. patent and trademark office (Tong & Jang, 2021; patent #11,030,487).

## Author summary

We explored how humans and artificial neural networks learn to recognize objects under noisy and ambiguous conditions, which is crucial for making sense of complex, real-world environments. Humans are naturally adept at identifying objects even when visibility is poor, like on a rainy or snowy day, or when objects are partially hidden. We wanted to ask if humans or neural networks receive training with very noisy images of objects, do they get better at the task? Also, if they are trained specifically with animate or inanimate object images, would recognition improve in general or only for the trained category? We found that humans became better at recognizing new object images in noisy conditions, but only for the categories they were trained on. Artificial networks initially struggled with noisy images but showed some general improvement from training, plus further benefits for the trained category. Interestingly, networks that were pre-trained to mimic the initial robustness of human observers only showed category-specific benefits of training, mirroring the effects of training in humans. Our findings highlight how humans adapt to challenging visual conditions, suggesting that learning plays an important role in understanding and navigating noisy, real-world settings.

## Introduction

Human object recognition is remarkably robust in real-world environments, even when sensory signals are weak, corrupted, or highly ambiguous, such as when one must drive through snow, rain, fog, or dimly lit conditions. Many studies have shown that people can successfully distinguish both simple and complex patterns in the presence of high levels of visual noise, blur, and other forms of image degradation [1–6]. Indeed, a signature property of human vision lies in its flexibility, adaptability, and overall capacity to generalize across diverse viewing conditions [7,8].

The robustness of human vision can be contrasted with the impressive yet brittle capabilities of current state-of-the-art deep neural networks (DNNs). The architectural design of convolutional neural networks was largely inspired by the feedforward organization of the primate visual system [9,10], including its repeated operations of filtering, non-linear rectification, and spatial pooling [11,12]. When trained on large image datasets on tasks of object recognition, these DNNs can accurately classify real-world images [13–17]; moreover, the representations learned by these networks can be used to model or effectively predict neural responses in the visual cortex of both non-human primates and humans [18–25]. Although standard DNNs can accurately classify clear object images, they perform poorly when tested with noisy, blurry, or degraded images of objects [2,4,26,27] unless they receive augmented training with such image perturbations [2,4,28,29]. Moreover, standard DNNs perform poorly at predicting human neural responses to noisy or blurry object images due to their

lack of robustness [4,29]. Such findings imply that standard DNNs, trained on an exclusive diet of clearly photographed object images, sharply deviate from how the human brain processes visual information.

What accounts for the robustness of the human visual system and its ability to tolerate noisy or degraded inputs? One possibility is that the visual system's processing architecture itself provides a degree of stability, allowing for the enhancement of relevant information and the suppression of noise (e.g., [30,31]). This might resemble a biologically evolved circuit that can readily perform image processing functions akin to filtering or denoising [32]. However, an alternative possibility is that robust vision is acquired from extensive experience, in which perceptual learning serves as the key mechanism for attaining robustness. How might one decide among these two competing accounts?

Given that sighted observers, even young children, have already encountered massive amounts of visual information in their lifetimes, it is challenging to conceive of how one might fully distinguish between the nativist versus empiricist accounts described above. However, if perceptual learning is indeed important for the acquisition of robust vision, then one would expect that adult observers should be able to acquire greater robustness to noisy stimuli if provided with adequate training. Indeed, if observers can improve after just a few hours of perceptual training, such that they are better at recognizing new noisy objects at test, then this would positively demonstrate that the human visual system can learn more robust representations to support successful generalization to novel stimuli. While prior work on perceptual learning has demonstrated improved discrimination of both simple and complex patterns in visual noise, these studies relied on the same sets of stimuli across training and test, so it remains unclear as to whether the learned representations were highly stereotyped or more flexible and robust [1,3,33,34].

The goal of our study was two-fold: to determine whether perceptual learning provides a viable account of how human observers acquire robustness to visual noise, and to evaluate whether DNNs can serve as viable models of human perceptual learning for complex tasks of object recognition. Observers had to identify objects presented in pixelated Gaussian noise by performing a 16-alternative classification task involving 8 animate and 8 inanimate categories drawn from the ImageNet database [35]. In previous work, we have shown that this set of object images can be used to effectively quantify how recognition accuracy varies as a function of noise level in both human observers and DNNs [4].

In the first experiment, we measured the threshold noise level at which objects could be successfully recognized in pixelated Gaussian noise, both before and after perceptual training, by presenting a novel object image in noise on every test trial. A separate set of 800 object stimuli (50 per category) were used for perceptual training, and both humans and DNNs were trained using the same set of object images. We found that human observers and DNN models showed significant benefits of training that generalized to the novel objects shown at test, with DNNs exhibiting much greater improvement due to their lack of robustness prior to training.

In a second experiment, we evaluated whether these perceptual improvements arose from category-specific or category-general benefits of learning, by training the humans and DNNs on noisy images drawn from the set of 8 animate categories or 8 inanimate categories and subsequently testing with all 16 categories. Our rationale for comparing animate versus inanimate training was motivated by the fact that these distinct superordinate categories are known to differ in their mid-level visual properties [22] and to activate well-differentiated representations in the human ventral visual system [36–41]. Thus, if training were to induce perceptually specific changes in the visual system, corresponding to the properties of the trained objects rather than the background noise, then one should be better able to detect effects of category-specific learning if the trained and untrained categories share less visual overlap.

Here, we found that standard pre-trained DNNs showed both category-general improvement and an additional category-specific benefit of training, whereas human observers only showed category-specific improvement. However, when we further evaluated DNN models that were pre-trained with moderately noisy images to match human pre-training accuracy, these models only showed category-specific improvement, and thereby matched the overall pattern of learning exhibited by human observers. A layer-wise analysis of DNN responses revealed that category-general learning effects emerged in the lower layers, whereas category-specific improvements emerged in much higher layers that represent more complex object information.

Taken together, our findings suggest that one component of visual robustness involves the learning of appropriately tuned representations at lower levels of visual representation. However, once these representations are acquired, additional training with noisy object examples leads to the fine-tuning of high-level representations, which while category-specific, are also sufficiently flexible to allow for successful generalization to novel exemplars from that category.

## Results

In this study, we developed a learning protocol that recruited human observers to perform object recognition tasks under noisy conditions over four training sessions. The training set consisted of 800 images obtained from the ImageNet validation set [35], drawing from 8 animate categories and 8 inanimate categories. The images were degraded by reducing their relative contrast and adding pixelated Gaussian noise, with the object's signal strength specified by the signal-to-signal-plus-noise ratio (SSNR) [4]. The training task involved presenting a new training image on each trial that gradually increased from an SSNR value of 0 (i.e., pure noise) to a maximum possible level of 1 (i.e., clear image), as illustrated in Fig 1. Observers were instructed to press a key to stop the animation sequence as soon as they felt confident enough to make a 16-alternative classification response. After that, they used a mouse-controlled virtual paintbrush to annotate the portions of the noisy image that informed their decision. Both before and after training, we used an adaptive staircase procedure [42] to estimate the threshold SSNR level for which each observer could reliably identify brief presentations of novel test images. All DNN models were trained and tested on the same sets of images as the human observers.

As can be seen in Fig 2A, human participants made their classification responses at progressively lower SSNR levels across the 4 training sessions, whereas mean classification accuracy remained consistently high (90%, 90%, 91%, and 90% for sessions 1–4, respectively). These changes in SSNR decision thresholds suggested that the observers could recognize the objects in progressively higher levels of noise over the course of training. A direct comparison of pre- versus post-training SSNR thresholds on the forced-choice adaptive staircase task confirmed that the observers did indeed improve in their robustness to noise after training (see Fig 2B). The fact that our training protocol led to successful generalization to novel test images indicated that the observers had acquired more robust representations that extended beyond stereotyped image-specific knowledge.

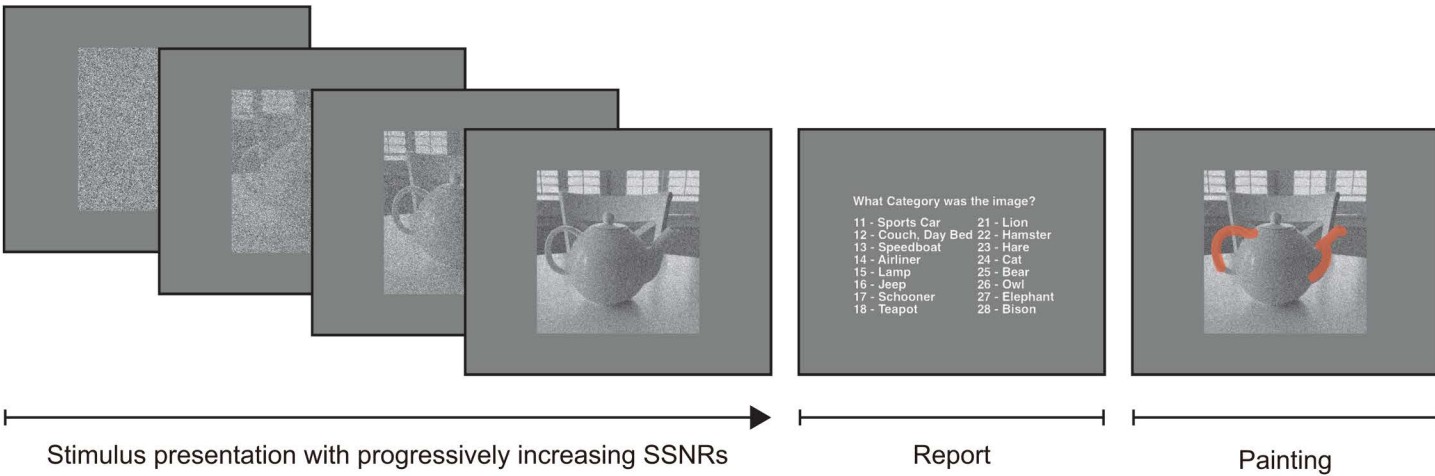

**Fig 1. Illustration of training protocol.** Example of a training trial with a teapot image gradually increasing in SSNR level until the observer chooses to make a 16-alternative classification response. The 16 possible categories included animate basic-level categories (bear, bison, elephant, hamster, hare, lion, owl, tabby cat) and inanimate basic-level categories (airliner, couch, jeep, schooner, speedboat, sports car, table lamp, teapot). SSNR levels of 0, 0.2, 0.4 and 0.6 are shown. After the classification response, the observer uses a virtual paintbrush to indicate the image regions most informative for the classification decision. Teapot image by Frank Tong.

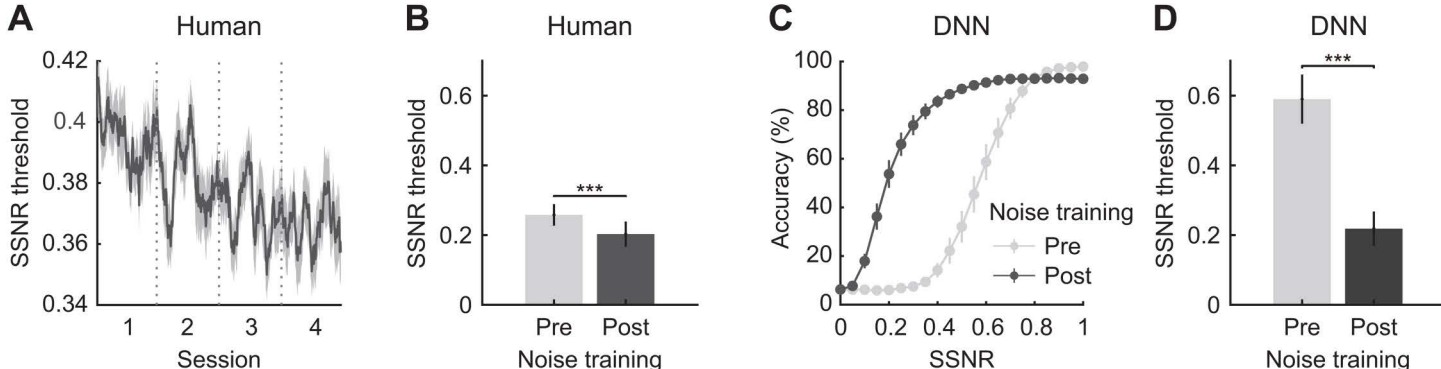

**Fig 2. Effects of training on SSNR thresholds in Experiment 1.** (A) Averaged human SSNR threshold estimates plotted over time across the four training sessions in Experiment 1 (N = 20). Data were smoothed using a 20-trial moving window. Shaded regions represent ±1 standard error of the mean (SEM). (B) Average human SSNR thresholds measured using an adaptive staircase procedure before training (light gray) and after training (dark gray). All error bars depict ±1 SEM. (C) Average classification accuracy for 6 DNN models plotted as a function of SSNR level, before noise training (light gray) and after training (dark gray). (D) Average SSNR thresholds for 6 DNN models with bar plot showing pre-trained (light gray) and post-trained (dark gray) performance. ***p < 0.001.

For comparison, we evaluated the performance of 6 convolutional neural network (CNN) models that were pre-trained on noise-free images using 1000 object categories from ImageNet [13–17], and then received additional training with noisy versions of the 800 training images (with SSNR ranging from 0.2 to 1). The novel test images could then be presented to the DNNs across a range of SSNR levels to characterize the SSNR threshold of each DNN model (Fig 2C). As expected, the standard pre-trained DNN models lacked robustness to noise and exhibited much higher SSNR thresholds than the human observers, consistent with prior work [2,4]. However, after training with the noisy training images, the DNNs showed dramatic improvement in their ability to recognize novel test images, with SSNR thresholds that were comparable to that of human observers (cf. Fig 2B and 2D). Statistical analyses indicated that the noise-trained DNNs exhibited lower SSNR thresholds than did the human observers before training (t(24) = 2.39, p = 0.025), while the two groups did not significantly differ after training (t(24) = 0.86, p = 0.40). These findings are quite notable, as we have previously shown that noise-trained DNN models significantly outperform human observers at recognizing objects in visual noise [4], though in that prior work, the human observers did not have the opportunity to receive additional training in the lab prior to test.

While the results of our first experiment indicated that both humans and DNNs can learn to become more robust to visual noise, we sought to clarify what types of information are acquired from such training. One possibility is that our perceptual learning task led to improvements in noise filtering or in the ability to discount the effects of pixelated Gaussian noise [1,43]. Alternatively, it is possible that the humans and DNNs acquired some form of knowledge regarding how candidate objects from the trained category can appear in extreme levels of visual noise [3]. To distinguish among these possibilities, we conducted a follow-up experiment that recruited two groups of human observers to receive training with either animate or inanimate objects in visual noise, using the same protocol as Experiment 1. Likewise, we trained two new sets of DNN models with noisy animate or inanimate objects.

As can be seen in Fig 3A, human observers showed significant improvements in the SSNR level at which they chose to make their classification responses in the training task, while maintaining similar levels of accuracy across sessions (95%, 95%, and 95% for each of the three sessions, respectively; within-subjects ANOVA, F(2, 62) = 0.104, p = 0.90). Comparisons of pre- versus post-training SSNR thresholds revealed that training led to category-specific improvements for both groups (Fig 3B). Observers trained on animate categories exhibited improved robustness when tested on animate categories (trend observed in 16/16 participants, t(15) = 8.23, p < 0.0001), but this improvement did not extend to inanimate

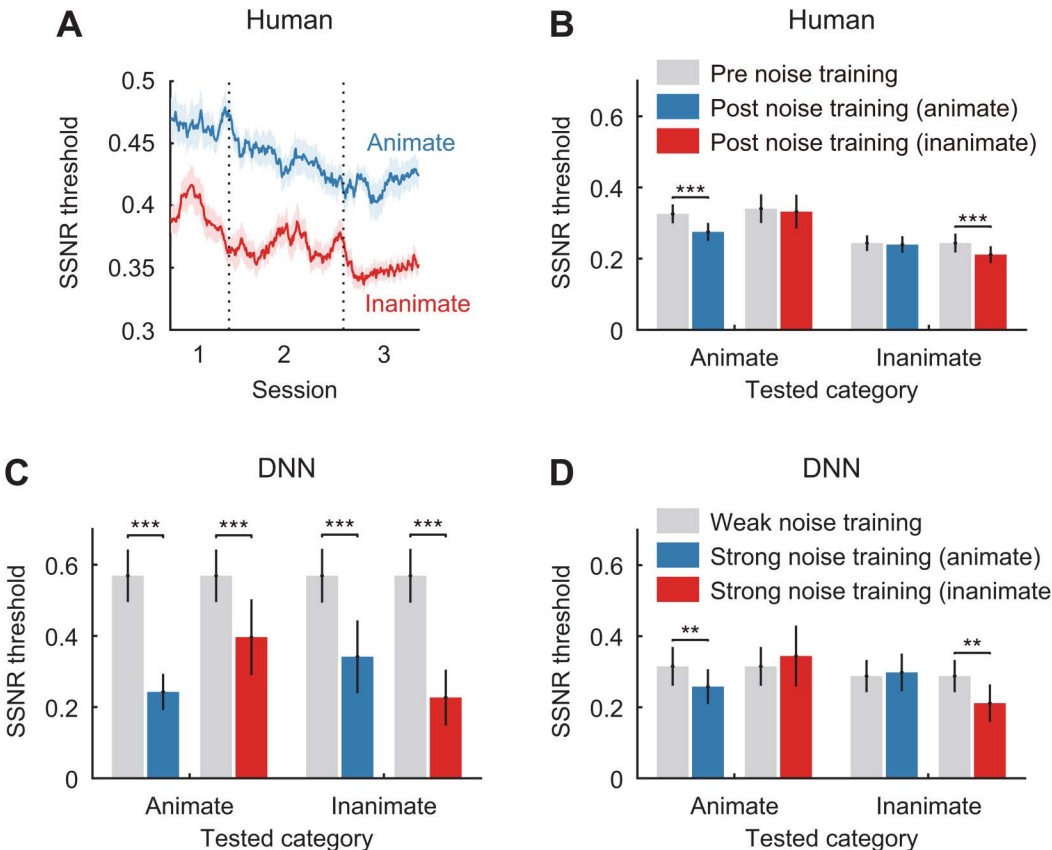

**Fig 3. Effects of training on SSNR thresholds in Experiment 2.** (A) Averaged human SSNR threshold estimates obtained during training with images of animate (blue) or inanimate (red) objects over three sessions in Experiment 2. Data were smoothed using a 20-trial moving window. (B) Average human SSNR thresholds obtained before noise training (gray bars) and after training with noisy animate (blue) or inanimate (red) objects. (C) Average SSNR thresholds for 6 DNN models obtained before training (gray bars), and after training with noisy animate (blue) or inanimate (red) object images. (D) Average SSNR thresholds for 6 DNN models obtained following weak noise training (gray bars), and after category-specific training with stronger noisy animate (blue) or inanimate (red) object images. **$p < 0.01$, ***$p < 0.001$.

categories ($t(15) = 0.81$, $p = 0.43$), as was further supported by a significant statistical interaction effect (within-subjects ANOVA, $F(1, 15) = 26.4$, $p = 0.0001$). Similarly, the group trained on inanimate categories showed significant improvement for inanimate categories (13/16 participants, $t(15) = 4.30$, $p = 0.0006$), while no improvement was seen for animate categories ($t(15) = 0.59$, $p = 0.56$); here, we observed a marginally significant interaction effect ($F(1, 15) = 3.09$, $p = 0.099$). Taken together, our findings indicate that humans trained with noisy object examples acquire category-specific knowledge of how candidate objects can appear in visual noise. While in some sense these learned representations are visually specific, they must be sufficiently flexible and robust to allow for successful generalization to novel, heterogenous test images.

Whereas the standard pre-trained DNN models lacked robustness, we found that category-specific training led to moderate improvement in threshold SSNR levels for the untrained category and substantial improvement for the trained category (Fig 3C). Similar to Experiment 1, the post-training thresholds exhibited by DNNs for their trained category were quite comparable to those of their human counterparts (animate objects, $t(20) = 2.02$, $p = 0.057$; inanimate objects, $t(20) = 0.74$, $p = 0.47$). In comparison, SSNR thresholds were considerably higher for the untrained category (animate objects, $t(5) = 5.63$, $p = 0.0024$; inanimate objects, $t(5) = 8.40$, $p = 0.0004$), but still much improved when compared to performance before noise training.

One reason why the human observers were more robust than the standard DNNs before training may be because humans have encountered many real-world examples of noisy environments (e.g., rain, snow, dust), whereas standard pre-trained DNNs lack such experiences [4]. To evaluate whether DNN models can more closely approximate the patterns of learning exhibited by human observers, we provided additional pre-training to the DNNs by presenting both animate and inanimate objects in moderate levels of noise (SSNR range from 0.6 to 1) until the networks performed at a level comparable to human pre-training accuracy. Next, we evaluated the effects of providing additional training on animate or inanimate objects embedded in higher levels of noise (SSNR range from 0.2 to 1).

As can be seen in Fig 3D, additional pre-training with moderate noise allowed the DNNs to match the pre-trained performance of human observers. More importantly, subsequent category-specific training allowed the DNNs to attain greater robustness to noise specific to the trained category. This proved true for DNNs trained on noisy animate objects (within-DNN ANOVA, $F(1, 5) = 17.51$, $p = 0.0086$ for interaction effect), and also for those trained on noisy inanimate objects (within-DNNs ANOVA, $F(1, 5) = 17.27$, $p = 0.0089$ for interaction effect). A comparison of the effects of category-specific training for humans and these DNNs revealed very similar effects of perceptual learning for both groups (cf. Fig 3B and 3D). These findings provide considerable support for the notion that DNNs can serve as effective models of human perceptual learning in tasks involving complex object recognition.

To assess how perceptual training led to changes in the visual representations of the DNN models, we performed a layer-wise analysis of susceptibility to noise across successive stages of processing. This analysis involved evaluating the correlational similarity (i.e., Pearson R) between activity patterns evoked by noise-free images and those same images presented at varying SSNR levels. We could then identify the threshold SSNR level required to attain a correlation similarity of 0.5 between these response patterns. For this analysis, lower SSNR thresholds indicate greater robustness to noise.

For standard pre-trained DNNs, selective training with either animate or inanimate stimuli led to a pronounced decrease in SSNR thresholds in the lower and intermediate layers for all categories; these results are shown for standard VGG-19 (Fig 4A). When compared with pre-training performance, we observed category-general benefits of training that increased in magnitude across layers 1–4, then remained stable throughout the intermediate layers (approx. layers 5–12), after which the SSNR thresholds for trained and untrained categories sharply diverged in the higher layers. In these higher layers (approx. layer 13 and above), one could observe a continuing trend of decreasing SSNR thresholds for the trained category, indicating computations akin to denoising across successive processing stages (see also [4]). In contrast, SSNR thresholds for the untrained category increased considerably in these higher layers, indicating a loss of noise robustness relative to that found in the intermediate layers. Taken together, we can conclude that training with either noisy animate or inanimate objects led to category-general gains in noise robustness in the lower layers of VGG-19, sustained benefits in the intermediate layers, and a divergence in robustness that selectively favored the trained category in the higher layers.

For DNNs that received additional pre-training with objects in moderate levels of noise, inspection of VGG-19's performance after pre-training revealed lower SSNR thresholds overall (gray curves) with a trend of decreasing noise susceptibility across successive layers (Fig 4B). Relative to this baseline, additional category-specific training led to a subtle decrease in SSNR thresholds across layers 1–12 for the untrained category and a more prominent decrease for the trained category. However, in the higher layers of these DNNs, the benefits of category-specific training gradually disappeared for the untrained category, while they were maintained for the trained category.

To further test the hypothesis that category-general and category-specific benefits of training are predisposed to emerge at different levels of visual processing, we evaluated the impact of freezing either the higher or lower layers of VGG19. First, motivated by the fact that category-general benefits of noisy object training grew considerably in magnitude across lower layers 1–4 (Fig 4A), we tested the impact of freezing layers 5–19, so that weight updates were restricted to these layers when performing stochastic gradient descent. A layer-wise analysis indicated that training with either noisy animate or inanimate objects both led to category-general improvements in SSNR threshold across layers 1–4 which persisted through the intermediate and higher convolutional layers (Fig 5A, left), while a modest amount category-specific

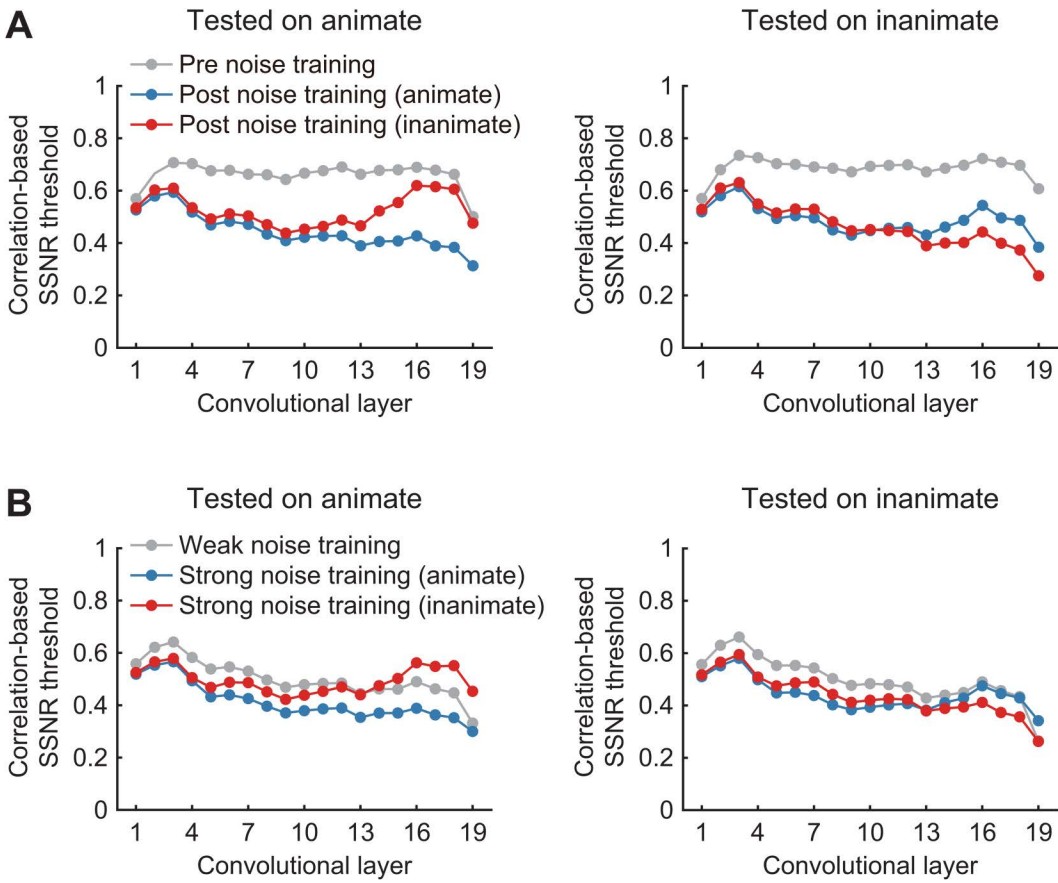

**Fig 4. Layer-specific analysis of noise susceptibility.** (A) Layer-wise analysis of stability of DNN responses to noisy test images for standard VGG19 before noise training (gray), and after training with noisy animate (blue) or inanimate (red) objects. Lower SSNR thresholds indicate greater robustness to noise. (B) The same layer-wise analysis performed on VGG19 after additional pre-training with objects in moderate noise (gray), followed by category-specific training with high levels of noise (blue, animate; red, inanimate).

divergence could be observed in the highest layers (i.e., fully connected layers 17–18 and classification layer 19). When SSNR thresholds were evaluated based on DNN classification output responses, we found that training led to strong category-general improvements in comparison to pre-trained VGG19, with a slight additional benefit observed for trained over untrained categories (Fig 5A, right). These results provide compelling evidence that the representations in the lowest layers of the DNN model are suitable for supporting category-general learning of noise robustness. (Additional DNN evaluations showed that if both lower and intermediate layers were kept trainable, such as layers 1–8, then layer-wise SSNR thresholds would tend to diverge in the higher layers for trained versus untrained categories, and classification SSNR thresholds also showed stronger category-specific differences (S1 Fig). Such findings are consistent with the fact that animate-inanimate distinctions are known to emerge in intermediate-layer representations in DNN models [22], so it can be assumed that the backpropagated error signal to these intermediate layers led to modified responses to the trained categories such that these differences persisted through to the downstream layers.)

We were also motivated by our prior finding that category-specific effects of noise training emerged in the higher layers of a fully trainable version of VGG19 (Fig 4B), which encouraged us to evaluate a network with learning restricted to these higher layers (13–19 trainable, with layers 1–12 frozen). A layer-wise analysis indicated that training led to decreases in SSNR threshold beyond layer 13, and more importantly, these benefits were primarily category-specific and tended to

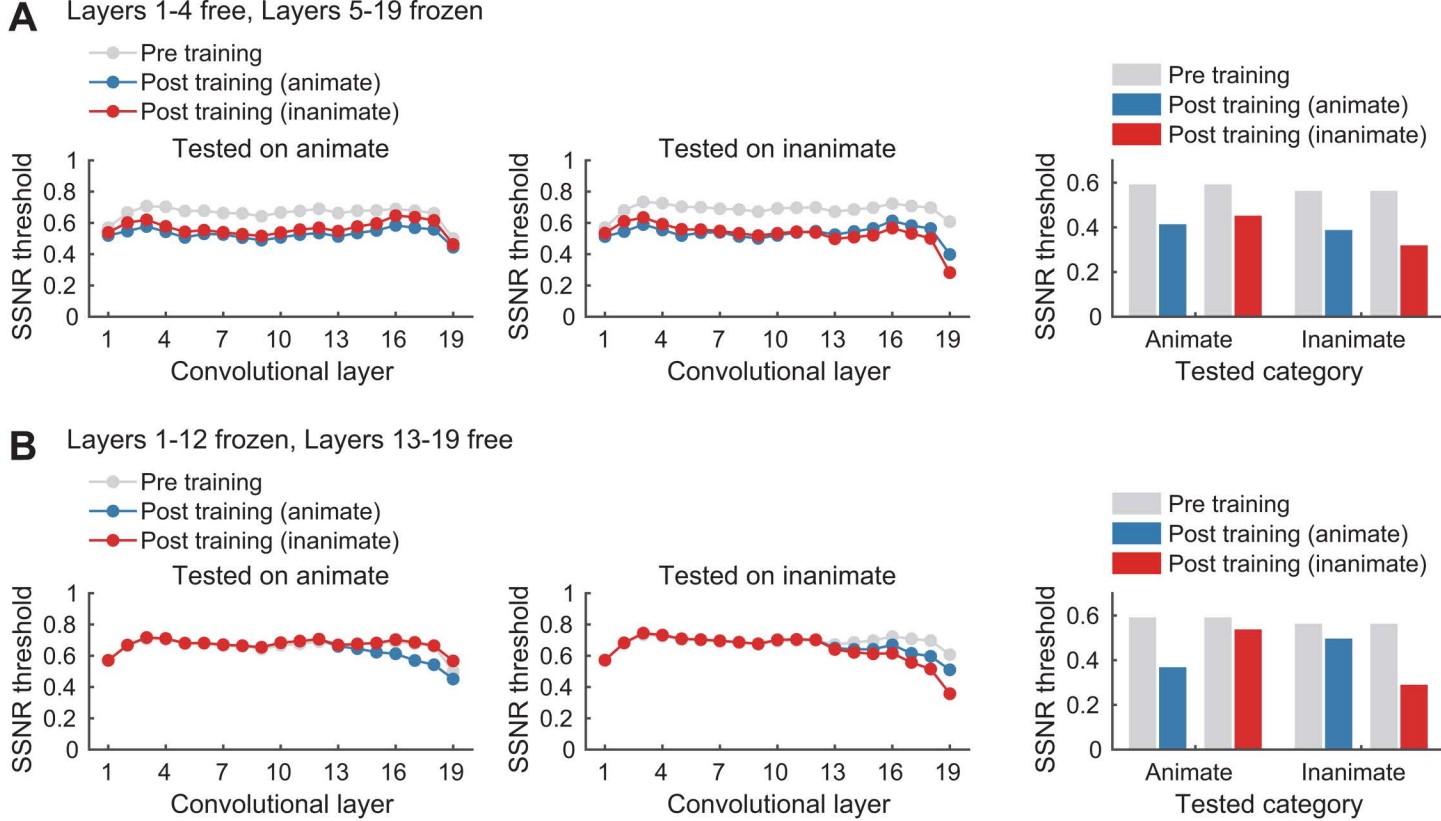

**Fig 5. Effects of layer-specific weight freezing on category-general and category-specific learning in VGG19.** (A) Left, Layer-wise noise susceptibility analysis of VGG19 when only the lower layers 1 to 4 were trainable (i.e., higher layers were frozen), assessed before training (gray) and after training with noisy animate (blue) or inanimate (red) objects. Note that to obtain strong category-general benefits of noise training, it was necessary to freeze both intermediate layers (5-12) and higher layers (13-19), otherwise greater category-specific learning would occur. Right, SSNR classification accuracy thresholds. (B) Same analysis as in (A), but with only higher layers 13 to 19 trainable and lower layers frozen. Results show that learning is mostly category-specific.

increase across successive processing stages (Fig 5B, left). Analysis of output responses revealed that training-induced improvements in classification SSNR thresholds were predominantly category-specific, with a marked benefit found for trained categories and minimal improvement found for the untrained categories (Fig 5B, right). Additional analyses revealed that if a greater proportion of intermediate layer representations were also made trainable along with the higher layers, then further improvements in SSNR threshold were observed for the trained category, while a modest amount of category-general improvement also began to emerge (S1B Fig, right). Overall, these results provide further support for the notion that category-specific improvement in noise robustness depends on higher-level DNN representations when compared with those that support category-general robustness.

Although some studies have reported that DNNs become overly specialized when trained with a specific type of visual noise, leading to failure with other types of noise [2], our previous work has suggested that partial generalization is possible [4]. In the present study, we were therefore curious to see how well our noise-trained DNN models would perform against multiple forms of image degradation drawn from the ImageNet-C benchmark [26]. As can be seen in Fig 6, DNNs trained with animate or inanimate objects in Gaussian noise were much more robust to other forms of noise (impulse, shot, speckle) as well as certain forms of image degradation (e.g., pixelate, JPEG compression); moreover, these benefits showed partial generalization to the untrained category. There were also suggestive trends of better performance for the

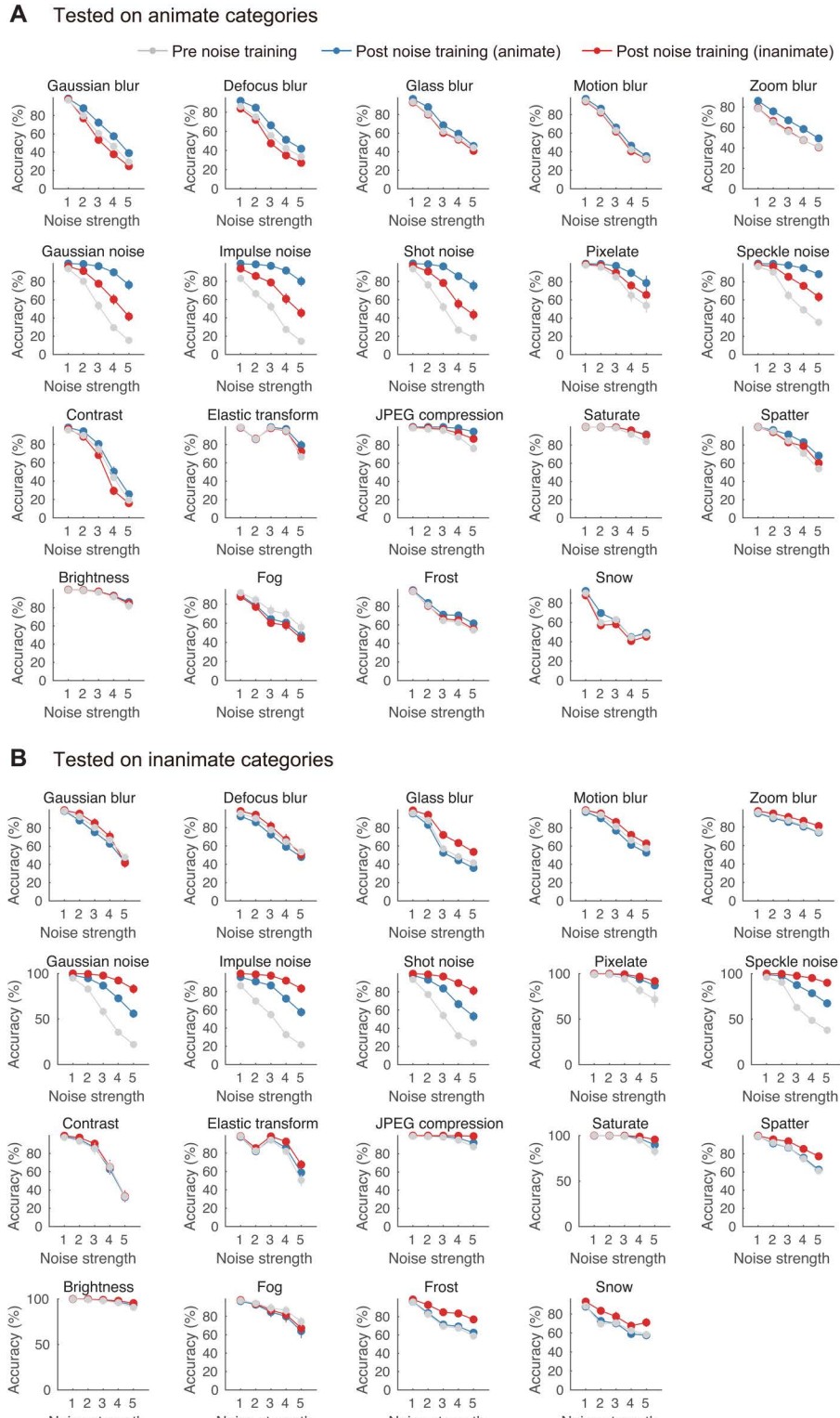

**Fig 6. Generalization to the ImageNet-C benchmark.** The average classification accuracy of 6 DNN models is shown before noise training (gray bars) and after training with noisy animate (blue) or inanimate (red) object images. Accuracy is plotted as a function of noise strength across 19 types of image degradation, with results shown separately for animate (A) and inanimate (B) object images. Error bars represent ±1 standard error of the mean.

trained category when dealing with certain types of blur, but such effects appeared to be quite small and variable across animate and inanimate test categories. Overall, the results of this analysis indicate that training with Gaussian noise leads to strong generalization to other types of noise that share a similar visual (if not statistical) resemblance, but these benefits do not extend to most of the image degradations that comprise ImageNet-C.

## General discussion

Our study evaluated whether perceptual learning can lead to the acquisition of robustness to visual noise, both in human observers and DNNs. We obtained novel evidence that human object recognition can indeed become more robust to severe levels of noise, such that the learned representations are tuned to the trained category but also sufficiently flexible to generalize to entirely novel test images. These findings provide support for our hypothesis that the robustness of the human visual system is likely acquired via mechanisms of perceptual learning.

Unlike human observers, standard DNNs lacked robustness to these image corruptions, consistent with prior work from our lab [4] and others [2,26]. However, after training with noisy animate or inanimate objects, the DNN models showed moderate gains in robustness for the untrained category; moreover, they were able to match human-level robustness for noisy test objects from the trained category. Thus, standard DNNs showed both category-general improvement and an additional category-specific benefit of training with noisy object images.

Given that human observers must deal with many forms of real-world noise, as can arise from encounters with rain, snow, fog, or other noisy elements, it is perhaps not surprising that recognition performance was already quite robust before our participants underwent training in the lab. When the DNN models were provided with additional pre-training at moderate levels of noise, we found that subsequent training with noisy animate or inanimate objects resulted in category-specific improvements that closely matched the patterns of learning exhibited by human observers. These findings demonstrate that DNNs can be effectively used to model and to predict how humans will improve at a complex visual task when given opportunities for learning.

A layer-wise analysis of DNN responses provided further insights into the effects of training on information processing across the successive stages of these models. Whereas training with noisy animate or inanimate objects led to category-general improvements in the lower layers of VGG-19, category-specific benefits became more pronounced in the higher layers. Similar patterns of results were observed in networks pre-trained with moderate levels of noise, where category-specific training benefits were concentrated in the higher layers of the model. These findings were further supported by a series of layer-freezing experiments. When learning was restricted to the lowest layers of the DNN (i.e., layers 1–4), we mostly observed category-general improvements in noise robustness. By contrast, when learning was restricted to the higher layers (e.g., layers 13–19), category-general benefits disappeared and the DNN mostly showed category-specific improvements in SSNR threshold. These computational results strongly suggest that category-specific acquisition of noise robustness depends on learning that occurs at higher levels of visual representation.

Extrapolating from these modeling results to human observers, it can be inferred that our training protocol led to the refinement of higher-level category-specific representations in the human visual system. In comparison, low-level feature representations did not seem to improve from this training regime, otherwise human performance for the untrained category should also have improved to a considerable extent. Presumably, for our observers, these low-level representations were already well-tuned and quite robust to noise before training.

In this study, we compared perceptual training with animate versus inanimate categories of objects, motivated by the fact that these distinct superordinate categories [44] are known to engage well-differentiated representations in the human ventral visual system (e.g., [22,36–41]). While our results provide compelling positive evidence that training with noisy objects does indeed lead to category-specific learning in humans, it remains an open question as to whether such specificity would occur at finer scales among basic-level categories with greater shared similarity. For example, would training with one set of animal categories lead to successful generalization to other untrained animal categories, given

that animals tend to share some overlap in their shape and textural properties? While the possibility of addressing this question in human participants extended well beyond the scope of our planned study, it was possible for us to explore this issue by training a new set of DNNs on a subset of animate or inanimate categories. Since we were interested in whether category specificity might be maintained at this finer level of granularity, we evaluated the performance of DNNs that initially received weak noise training with all 16 object categories, followed by strong noise training with only 4 of the basic-level categories drawn from either the animate or inanimate object sets. The results of this exploratory analysis revealed that training led to category-specific improvements at the basic level in 3 out of 4 cases, while we did also observe that DNNs trained with animate set 1 exhibited better generalization performance for animate set 2 (see S2 Fig). These findings suggest that perceptual training with noisy objects can lead to category-specific improvements at the basic level, although a certain degree of generalization can also occur at the broader superordinate level. Future studies with human observers could explore whether similar patterns of perceptual training emerge, although such investigations would likely require substantially larger sample sizes to detect potentially subtle generalization effects with adequate statistical power.

The learning benefits found in our study further demonstrate how human robustness to visual noise can be further improved by only a few hours of training in a laboratory setting, with 800 training trials in Experiment 1 or 400 trials in Experiment 2. Although Experiment 2 revealed these improvements to be category-specific, the acquired information was sufficiently flexible and robust to support successful generalization to novel real-world object images, thereby indicating a form of high-level visual learning. Likewise, it is notable that our DNN models showed improved robustness to noise after training with the same set of object images used for training in the human studies. Although it is commonly believed that DNN models require extremely large datasets to learn appropriate visual representations, we found that additional fine-tuning with only a few hundred noisy object images could lead to major improvements in DNN model robustness. If one assumes that DNNs can provide an informative measure of how much people might possibly learn in a given situation or from a limited set of training examples, then DNN models could prove useful for providing new insights into human perceptual learning.

Taken together, our human behavioral results, in conjunction with the DNN modeling results, provide strong support for our hypothesis that a limited number of encounters with highly challenging viewing conditions can provide the basis for the acquisition of visual robustness. Such conclusions also receive support from anecdotal reports by people who move to a much colder climate with snowy winters. While the first few occasions of driving through a snowstorm can be a terrifying white-knuckled experience, most individuals improve and become seasoned drivers after just one or two winters. Thus, even in adulthood, the human visual system appears capable of considerable plasticity, allowing for rapid and flexible learning, as was demonstrated by the participants in our study.

Our training paradigm, which involved presenting a novel object image on every trial, can be contrasted with traditional approaches to perceptual learning. These studies have typically focused on perceptual training with simple low-level visual features [34,45,46] or with a limited number of complex stimuli that are used for both training and test (e.g., [3]). In the absence of testing for generalization to novel stimulus conditions, it can be difficult to discern whether improvements in performance are due to the modification of low-level feature representations or higher-level object representations. Here, by evaluating human performance using heterogeneous and unpredictable object images, we could ensure that recognition performance had to rely on flexible, higher-level visual representations.

So far, only a few studies have used DNN models to characterize human perceptual learning, primarily focusing on tasks of orientation discrimination. These studies have found that DNNs are sensitive to the effects of training on high versus lower precision tasks, training with single or multiple spatial frequencies, and can help account for effects of generalization to new orientations, new spatial frequencies or other stimulus conditions [47–49]. Our study expands on this approach to characterize how humans and DNNs improve at complex tasks of object recognition that require dealing with ambiguity, noise, and stimulus uncertainty. While DNN models are only beginning to be introduced into studies of perceptual learning, we believe that this computational approach is highly promising. Indeed, convolutional neural network

models are inspired by the feedforward architecture of the human visual system, and could help provide a computational framework for understanding the bases of many forms of visual learning.

While this study primarily focused on robustness to Gaussian noise, another important question concerns how robustness is acquired for multiple forms of image degradation. Some research has suggested that DNNs trained with one type of visual noise are unable to generalize to other types of noise (e.g., uniform vs. salt-and-pepper noise) [2], while other studies have reported partial generalization success [4]. Here, we found that DNNs trained with objects in Gaussian noise could generalize well to the impulse, shot and speckle noise conditions from ImageNet-C; however, such training only led to slight improvements in the ability to recognize blurred objects. These findings are consistent with the proposal that the generalization performance of DNN models may be linked to the spatial frequency content of the noise components used for training and test; if both have relatively greater power at high spatial frequencies, that could allow for a certain degree of generalization as those frequency components are down-weighted [50]. In other recent work, we have shown that DNNs trained with a mixture of clear and blurry images become more robust to both blur and noise, with significant improvements for 14 out of 19 ImageNet-C conditions [29]. Thus, when compared with the current findings we see evidence of an asymmetry: while blur training enhances robustness to Gaussian noise [29], Gaussian noise training does not confer the same level of robustness to blur (Fig 6). This discrepancy may highlight differences in the types of visual information that are learned from distinct image distortions. Further studies could reveal whether strategic exposure to a variety of visual distortions might enhance DNN robustness to a broader array of image challenges.

In the present study, we found that feedforward CNN models can capture important aspects of human perceptual learning when trained on a highly challenging task of recognizing objects in severe levels of visual noise. However, it is worth noting that standard deep neural networks typically lack top-down feedback and recurrent mechanisms—network properties that have been shown to markedly enhance visual recognition under complex, degraded visual conditions [51–54]. Moreover, perceptual learning has been found to modulate how top-down processing occurs in the visual system, thereby influencing how the brain integrates sensory input with prior knowledge [55–57]. Of considerable relevance, a recent study found that adding a top-down reconstructive attentional mechanism to DNN models led to improved recognition of MNIST digits in noise; moreover, this approach was successfully extended to classify objects in the presence of various forms of image degradation [58]. Incorporating such recurrent or top-down mechanisms into DNN architectures might not only improve their robustness, but further has the potential to provide insights into the mechanisms that underlie the robust nature of human vision.

In conclusion, the present study underscores the complex interplay between category-specific and category-general mechanisms in the perceptual learning of noise-robust visual representations. Through a systematic comparison of humans and DNN models, we provide key insights into the adaptive nature of visual learning in both systems. Our findings not only enhance current understanding of how the human visual system processes noisy or ambiguous visual inputs, they also provide guidance for how artificial vision systems can be made more robust and better aligned with human perceptual capabilities.

## Methods

### Ethics statement

This study was approved by the Institutional Review Board of Vanderbilt University (IRB #040945 "Studies of Human Visual Processing"). All participants provided written informed consent prior to participation and reported having normal or corrected-to-normal visual acuity. In Experiment 1, 20 participants were recruited, while Experiment 2 involved a separate group of 32 participants. Participants received either monetary compensation or course credit with supplementary monetary compensation for taking part.

## Visual stimuli

Our training object images were obtained from the ImageNet validation set [35], and consisted of 50 images drawn from each of 16 categories. The stimulus categories included both animate objects (bear, bison, elephant, hamster, hare, lion, owl, tabby cat) and inanimate objects (airliner, couch, jeep, schooner, speedboat, sports car, table lamp, teapot). All images were converted to grayscale to eliminate color cues for recognition.

The level of noise in the images was manipulated by varying the signal-to-signal-plus-noise ratio (SSNR), defined as $w$ in the equation $T = w \cdot S + (1 - w) \cdot N$, where $S$ represents the source image, $N$ denotes the noise image, and $T$ is the target image with added noise [59]. The SSNR ranges from 0 (noise only) to 1 (signal only). The pixel intensities for $S$ and $N$ were scaled between 0 and 1. For the noise $N$, random values were sampled from a Gaussian distribution centered at 0.5 with a standard deviation of 1/6, ensuring 99.73% of pixel intensities remained within the 0–1 range. Any pixels outside this range were clipped accordingly.

## Experiment 1

In the first experiment, participants had to classify a novel training image embedded in noise on every trial, performing a total of 800 trials across 4 sessions, with each session lasting approximately one hour. On each trial, participants viewed an image where a target object gradually emerged from noise as the SSNR level increased from 0 to 1 in 0.025 increments every 400ms. Participants pressed a key to halt this animation sequence as soon as they were ready to make a sixteen-alternative forced-choice classification response. Additionally, participants used a virtual paintbrush controlled by a mouse pointer to annotate the portions of the noisy image that were most informative for their decision. To incentivize both fast and accurate responses, participants received a small monetary bonus for correct responses based on the speed of their response, and received feedback at the end of each trial. Prior to the first training session, participants completed 16 practice trials with a different set of novel images to familiarize themselves with the procedure.

Both before and after the training trials, we measured the threshold SSNR level at which participants could successfully classify noisy test objects that were briefly presented for 200ms. Specifically, we used an adaptive staircase procedure called QUEST [42] to estimate each observer's recognition accuracy by SSNR curve, which was updated on a trial-by-trial basis to determine the SSNR level of the next test stimulus to present. Noise robustness was measured by the SSNR threshold level corresponding to a 57% performance criterion (inflection point) in a 48-trial run, with initial threshold and standard deviation estimates set at 0.5 and 0.4, respectively. For these pre-training and post-training test measures, we used 96 object images (6 per category) not included in the training sessions. Four independent raters, including one of the authors, determined the noise thresholds for all 96 images using our training protocol but without the annotation task. Based on these subjectively measured thresholds, two image sets (48 each) with closely matched difficulty levels were created, and these were assigned to serve as pre-training or post-training test stimuli, counterbalanced across participants. To minimize potential demand characteristics, participants were not informed that the entire study focused on testing for effects of perceptual learning.

## Experiment 2

In Experiment 2, participants were randomly assigned to undergo training with noisy object images drawn from either the 8 animate categories or the 8 inanimate categories. Each of the two groups performed 400 training trials across 3 sessions (100 trials in the first session, 175 trials in the second session, and 125 trials in the final session). The noise sensitivity (or noise robustness) of both groups was assessed both before and after training using the adaptive staircase procedure, with separate SSNR threshold estimates obtained for animate objects and inanimate objects. Our experimental design sought to compare the effects of training with animate versus inanimate object images, which are highly visually distinct, to enhance the likelihood of detecting category-specific effects of training should they exist. That said, our study was not designed to test for possible effects of generalization within a trained superordinate category, as one could

potentially test for generalization using visually similar categories (e.g., lion vs. puma) or visually dissimilar categories (e.g., lion vs. aardvark) and the potential problem space is vast.

To enhance the reliability of our adaptive staircase test, three raters evaluated the difficulty level of 768 individual images. These ratings were used to create 2 sets of animate object images of comparable difficulty, and 2 sets of inanimate objects with balanced difficulty (48 images per test set). The assignment of the image sets for pre- or post-training test was counterbalanced across participants. To ensure the consistency of training effects, most participants completed all three sessions within one week. Additionally, a criterion for inclusion was established: participants who did not achieve at least 90% accuracy in the first session were excluded from the remaining sessions.

## Convolutional neural networks

We evaluated multiple trained versions of six CNN models: AlexNet [14], VGG16, VGG19 [15], GoogLeNet [16], ResNet50 [13], and Inception-v3 [17]. All modeling experiments were implemented using PyTorch. Each network was initialized with pre-trained weights from ImageNet, and noised-trained models were further trained on objects in visual noise using the same set of 800 training images that the human observers viewed in Experiment 1 or only 400 training images (i.e., animate or inanimate) for Experiment 2. During training, the input images were degraded with varying levels of Gaussian noise [4,59], with the specified signal-to-signal-plus-noise ratio randomly sampled from a uniform distribution from 0.2 to 1.0. In Experiment 2, a matching number of clear images from the untrained categories were concurrently used for training. To ensure that data augmentation did not introduce unintended interference with noise robustness, we applied only minimal augmentation techniques, specifically RandomResizedCrop and RandomHorizontalFlip, using their default parameters in PyTorch. (Methods such as color jitter were not used.) Images were resized to 224 × 224 pixels, converted to grayscale, and transformed to RGB by replicating the values across all 3 color channels to ensure compatibility with the pre-trained models. Each image was normalized using the mean (0.449) and standard deviation (0.226) of the ImageNet training set. The networks were optimized using a stochastic gradient descent algorithm with a fixed learning rate of 0.01 and a weight decay of 0.0001 for 100 epochs. Control CNNs were trained on noise-free training images using identical procedures.

Unlike human participants, CNNs could be repeatedly tested with the same object image across a full range of SSNR levels to assess threshold performance. Thus, noise thresholds before and after training were estimated from the complete SSNR-accuracy curve using the same test images as were used in the human experiments. Since category-specific noise training in Experiment 2 might lead to biased classification decisions in favor of the trained category for noisy stimuli, we restricted CNN responses by using the highest confidence response from the relevant trained or untrained categories to estimate SSNR thresholds.

Additionally, to investigate whether initial weak noise training on both animate and inanimate categories would elicit similar effects to those observed in humans (specifically, the absence of category-general effects), we provided modest noise pre-training to a subset of models using the 800 training images with SSNRs ranging from 0.6 to 1.0, which included both animate and inanimate categories. This was followed by training with SSNRs ranging from 0.2 to 1.0 for either animate or inanimate categories.

## Layer-wise noise susceptibility analysis

For the layer-wise noise susceptibility analysis, we followed the methodology outlined by Jang et al. (2021) [4]. Specifically, we evaluated the consistency of activity patterns for objects with increasing noise levels by calculating the Pearson correlation coefficient between responses to noise-free images and the same images across varying SSNR levels. High robustness to noise was indicated by similar responses to both noisy and noise-free images. This analysis was conducted on all layers of VGG19, including 16 convolutional layers and 3 fully connected layers. The correlation-by-SSNR curve was rescaled to a range 0–1 to eliminate offsets at 0 SSNR. A logistic function was then fitted to the correlation-by-SSNR curve for each layer, and the SSNR level at which correlation strength reached 0.5 was identified as the SSNR threshold.

### Diagnostic feature analysis in humans and neural networks

To compare diagnostic image regions utilized by CNNs with those obtained from human participants, we applied layer-wise relevance propagation [60] on VGG19 model responses to noisy object images. This technique identifies diagnostic features that contribute to network predictions by decomposing the network's output into relevance scores, which are subsequently backpropagated to the input layer. To align with the resolution of human-drawn diagnostic regions, the relevance maps were smoothed using a Gaussian kernel with a standard deviation of 3 pixels. We assessed the similarity between the diagnostic regions of humans and CNNs by calculating Pearson correlation coefficients and overlap ratios. The specifics of parameter choices were based on Jang et al. (2021) [4]. Additionally, to evaluate human-to-human similarity in saliency maps, we randomly divided the data into two groups and computed the correlation and overlap ratios between the average maps from these groups. This sampling procedure was repeated 10 times, with the average values reported. The results of this additional analysis revealed that noise-trained VGG-19 performed better at predicting the diagnostic spatial regions reported by human observers, consistent with our prior work [4], and that spatial predictions were better for noisy objects from the trained category than the untrained category (see S3 Fig).

## Supporting information

**S1 Fig. Extended analysis of the effects of layer-specific weight freezing on training-induced changes in VGG19.** Separate versions of VGG19 received training with noisy animate or inanimate objects, with only a subset of layers that remained trainable. (A) Layer-wise noise susceptibility analysis (left) and output-level SSNR classification thresholds (right) when progressively more of the early and intermediate layers were trainable (i.e., layers 1–4, 1–8, 1–12, 1–16 from top to bottom). (B) Complementary analyses of VGG19 when progressively more of the higher layers were trainable (i.e., layers 17–19, 13–19, 9–19, 5–19).
(EPS)

**S2 Fig. Evaluation of DNNs trained with a subset of animate or inanimate categories on generalization performance.** Separate sets of initially weak-noise-trained DNNs were trained with one of four noisy object subsets: Animate1 (bison, hamster, lion, owl), Animate2 (bear, elephant, cat, hare), Inanimate1 (airliner, jeep, speedboat, table lamp) or Inanimate2 (sports car, couch, schooner, teapot). We then measured classification-based SSNR thresholds to test for specificity or generalization of training benefits within a superordinate category; 4AFC classification thresholds (with slightly higher accuracy criterion) were used to minimize potential effects of bias that might arise from subset training. Results are further compared with the DNNs trained on all 8 animate or inanimate categories. Training-induced changes in DNN performance for each test subset were evaluated using paired t-tests; *p < 0.05, **p < 0.01, ***p < 0.001.
(EPS)

**S3 Fig. Comparison of diagnostic features used by humans and DNNs.** Correlational similarity and overlap ratio of the spatial profile of diagnostic features reported by human observers and those measured in DNNs across a range of SSNR levels. Dashed lines indicate ceiling-level performance based on human-to-human correspondence.
(EPS)

## Author contributions

**Conceptualization:** Hojin Jang, Frank Tong.

**Data curation:** Hojin Jang.

**Formal analysis:** Hojin Jang.

**Funding acquisition:** Hojin Jang, Frank Tong.

**Investigation:** Hojin Jang.

**Methodology:** Hojin Jang, Frank Tong.

**Project administration:** Frank Tong.

**Supervision:** Frank Tong.

**Validation:** Hojin Jang.

**Visualization:** Hojin Jang.

**Writing – original draft:** Hojin Jang, Frank Tong.

**Writing – review & editing:** Hojin Jang, Frank Tong.

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
