## [Decision Letter · Decision Letter 0]

23 Feb 2025

PCOMPBIOL-D-24-01887

Category-specific perceptual learning of robust object recognition modelled using deep neural networks

PLOS Computational Biology

Dear Dr. Jang,

Thank you for submitting your manuscript to PLOS Computational Biology. After careful consideration, we feel that it has merit but does not fully meet PLOS Computational Biology's publication criteria as it currently stands. Therefore, we invite you to submit a revised version of the manuscript that addresses the points raised during the review process.

Please submit your revised manuscript within 60 days Apr 25 2025 11:59PM. If you will need more time than this to complete your revisions, please reply to this message or contact the journal office at ploscompbiol@plos.org. Please include the following items when submitting your revised manuscript:

We look forward to receiving your revised manuscript.

Kind regards,

Grace W. Lindsay

Guest Editor

PLOS Computational Biology

Hugues Berry

Section Editor

PLOS Computational Biology

**Journal Requirements:**

At this stage, the following Authors/Authors require contributions: Hojin Jang, and Frank Tong. Please ensure that the full contributions of each author are acknowledged in the "Add/Edit/Remove Authors" section of our submission form.

4) We notice that your supplementary Figure is included in the manuscript file. Please remove it and upload it with the file type 'Supporting Information'. Please ensure that each Supporting Information file has a legend listed in the manuscript after the references list.

Potential Copyright Issues:

i) Please confirm (a) that you are the photographer of 1, and S1A, or (b) provide written permission from the photographer to publish the photo(s) under our CC BY 4.0 license.

1) State the initials, alongside this funding source (National Research Foundation of Korea). For example: "This work was supported by the National Institutes of Health (####### to AM; ###### to CJ) and the National Science Foundation (###### to AM)."

2) State what role the funders took in the study. If the funders had no role in your study, please state: "The funders had no role in study design, data collection and analysis, decision to publish, or preparation of the manuscript.".

**Reviewers' comments:**

Reviewer's Responses to Questions

Reviewer #1: Two experiments explored the relationship between object recognition robustness and training on images of objects that were degraded by Gaussian noise. This was done for both human subjects and several well-studied CNN models to determine whether CNNs can be used to model the perceptual learning occurring in humans. In Experiment 1, the researchers found that both humans and CNNs showed significant benefits (lower thresholds) from noise training. These benefits were larger for the models, but the authors showed that this was due to the pretrained models having zero experience with noisy images (unlike humans). In Experiment 2, the researchers asked a similar question only now manipulated whether the category of the object was animate or inanimate to determine whether the training benefits are category general or category specific. They found that, for humans, there were training benefits only for the trained-on category (category-specific), whereas CNNs showed both category-general and category-specific learning. The researchers also showed that, once CNNs were pre-trained with noisy images, they too showed only category-specific training benefits (like humans). From layer analyses of the CNNs, they concluded that category-general learning is due to weight changes in the lower layers whereas category-specific learning is due to weight changes in the higher layers.

I appreciated the opportunity to read this paper. The research question is interesting, important, and as the authors state, relatively unstudied. The authors fill this gap with a solid study that answers some basic questions about the relationship between object recognition robustness and training on noise-degraded images. The data patterns are clear and well analyzed, and the conclusions do not overstep the results of these analyses. My only criticisms in reading the manuscript were in wanting the see the two very things that the researchers mention in future work (lines 376-386). Greedily, I would have liked to see these extensions in the current manuscript, but neither extension is easy and publication of the current work shouldn’t be delayed in the interim. When the authors do get around to these directions, they should consider the ImageNet-C dataset (Hendrycks D, Dietterich TG. Benchmarking Neural Network Robustness to Common Corruptions and Perturbations. ICLR 2019) that introduces a range of visual corruptions to ImageNet images. With respect to their goal of considering the role of recurrency in training, they may want to consider a recent paper that used recurrency to model top-down processing and showed it to improve object recognition robustness (Ahn, S., Adeli, H., & Zelinsky, G. J. The attentive reconstruction of objects facilitates robust object recognition. PLOS Computational Biology, 20(6) 2024).

Perhaps one small way that the manuscript could be strengthened is if a stronger connection were made to the attention literature. Humans get constant training on noisy images in the context of normal active vision and the blur/noise that is experienced in peripheral vision. An object selected for fixation is first perceived as blurred in the visual periphery, but the visual system then gets immediate feedback on its high-resolution version following the eye movement to that object. This is an even more specific form of noise training than in the researchers’ category-specific condition because the noise is embedded in the specific object to be recognized, not an object that is categorically related. It would be interesting to know how this object-specific training benefit would compare to the category-specific and category-general benefits reported here. My intuition is that this sort of training benefit would be captured by weights in the lower layers, but this would seem to break with the pattern reported here by the researchers, where category-specific benefits are associated with learning in the higher layers. If the researchers choose to make this connection in their introduction, they may want to look at studies that predict the movements of a foveated retina (e.g., Yang, Z., Mondal, S., Ahn, S., Xue, R., Zelinsky, G., Hoai, M., & Samaras, D. Unifying Top-down and Bottom-up Scanpath Prediction Using Transformers. CVPR 2024; Yang, Z., Huang, L., Chen, Y., Wei, Z., Ahn, S., Zelinsky, G., … & Hoai, M. Predicting Goal-directed Human Attention Using Inverse Reinforcement Learning. CVPR 2020), which is a related question.

Some small things.

Line 89. "fundamentally misaligned" seems a bit strong. Augmenting CNN training with noisy images easily fixed this problem, so it couldn’t be fundamental.

Lines 93-94. This suggestion of a biological circuit sounded to me a bit of a straw man. If a simple denoising circuit exists in the visual system, wouldn’t someone have found it by now? Is there evidence for such a circuit? And if there was a de-noising circuit, shouldn’t there be minimal to no perceptual learning, which is known not to be true? My point being, a data-driven approach based on perceptual learning and experience just seems far more plausible.

Line 233. Figure 3D is referenced in the text but is missing from the caption.

Signed,

Greg Zelinsky

Reviewer #2: In this paper, the authors measured human behavioral performance and DNN output to understand how training on noise image input would affect object recognition performance. They found in both humans and DNNs that training with noisy images improves performance on novel images not included in training and that training may affect performance in a category-specific manner.

Overall, I found the study to be cleverly designed and well conducted, and the results are clean and informative. The contrast between human and DNN performance is particularly informative and allows us to gain a deeper understanding of how training on noisy images may affect object recognition performance. I just have one comment that I would like the authors to consider.

The authors argue that training is category-specific. However, I wonder if training is, in fact, object-specific. To show that training is truly category-specific, training should only involve half of the animate or inanimate objects. If the effect of training is generalized to the untrained objects in the same animate or inanimate category, then we have support for the category specificity of training. The results as they stand right now, however, are also consistent with object-specific training: once the human observer or DNN learns to recognize a particular object in noise, they can generalize such learning to new exemplars of the same object; if training only involved, e.g., animate objects, then performance would only improve for those objects and not inanimate objects. This possibility should be addressed, perhaps with a new study. I believe that addressing this point thoroughly will make this study a more significant contribution to the field.

Reviewer #3: This paper presents a novel perspective on noise robustness in human object recognition compared to artificial neural networks (ANNs) pretrained on image classification. The study makes two key contributions: First, it demonstrates that humans' robust perception of noisy images partially stems from their exposure to visual noise during development. The authors show that pretraining ANNs on noisy natural images improves their robustness in object categorization. This suggests that the previously observed gap in robustness between humans and ANNs can be, at least partially, attributed to differences in training exposure. Second, they demonstrate that additional training with noisy images leads to category-specific improvements that don't generalize to other categories - a pattern observed in both humans and ANNs. This kind of side-by-side comparison of human and ANN learning processes offers insights into the strengths and limitations of neural network models while advancing our theoretical understanding of neural and behavioral aspects of learning in humans and other animals. For these reasons, and also the importance of presented results, I strongly support the publication of this paper. However, I have a few concerns that should be addressed:

Major Comments:

Choice of animate vs. inanimate Categories: While the brain shows distinct structural differences in processing animate versus inanimate objects, I'm not aware of similar systematic differences in ImageNet-trained ANNs. The authors should either provide evidence for this distinction in ANNs or justify their category choice. Why not use finer-grained object categories to demonstrate category-specific robustness? For instance, training on two object categories and testing robustness on a third? Does the degree of category-specificity depend on the choice of train/test categories? Empirical evidence showing the phenomenon's consistency across different category combinations would strengthen the findings.

Layer-specific effects: The authors should consider an ablation study to support their claims about layer-related effects on category-general and category-specific robustness. For example, freezing early layers during fine-tuning could verify whether this prevents category-general improvements.

Impact on noiseless image performance: How does training with noisy samples affect the model's performance on standard ImageNet images? Are there meaningful changes in classification accuracy?

Generalization of noise robustness: While exposure to noise during training may enhance robustness, it may not be the only contributing factor. Does training with Gaussian noise, as used in this study, improve robustness to other types of noise? How do early layer modifications affect performance on clean images? Do these layers respond similarly to different types of noise?

Minor Comments:

Please clarify how the choice of data augmentation techniques during model training influenced the robustness results.

For the SSNR threshold learning curves in Figures 2 and 3, please specify whether these represent individual or averaged subject data. What is the inter-subject variability? Does the observed effect (category-specific robustness) appear in individual subjects or only in group-level analyses?

**Have the authors made all data and (if applicable) computational code underlying the findings in their manuscript fully available?**

Reviewer #1: Yes

Reviewer #2: Yes

Reviewer #3: Yes

PLOS authors have the option to publish the peer review history of their article (what does this mean? ). If published, this will include your full peer review and any attached files.

**Do you want your identity to be public for this peer review?** For information about this choice, including consent withdrawal, please see our Privacy Policy .

Reviewer #1: **Yes: ** gregory zelinsky

Reviewer #2: No

Reviewer #3: No

**Figure resubmission:**
---

## [Decision Letter · Decision Letter 1]

27 May 2025

PCOMPBIOL-D-24-01887R1

Category-specific perceptual learning of robust object recognition modelled using deep neural networks

PLOS Computational Biology

Dear Dr. Jang,

Thank you for submitting your manuscript to PLOS Computational Biology. After careful consideration, we feel that it has merit but does not fully meet PLOS Computational Biology's publication criteria as it currently stands. Therefore, we invite you to submit a revised version of the manuscript that addresses the points raised during the review process.

Please submit your revised manuscript within 60 days Jul 27 2025 11:59PM. If you will need more time than this to complete your revisions, please reply to this message or contact the journal office at ploscompbiol@plos.org. Please include the following items when submitting your revised manuscript:

We look forward to receiving your revised manuscript.

Kind regards,

Grace W. Lindsay

Guest Editor

PLOS Computational Biology

Hugues Berry

Section Editor

PLOS Computational Biology

**Additional Editor Comments :**

Please note that two of the reviewers still have substantial concerns about the extent to which certain claims are supported by the evidence. These need to be addressed either through further analysis or a re-framing of the results to ensure that all claims are fully supported.

**Reviewers' comments:**

Reviewer's Responses to Questions

Reviewer #1: The researchers satisfied all of my concerns.

Reviewer #2: I am not sure the authors convincingly addressed my comment. To argue that the training effect is specific to the animate or inanimate distinction at the superordinate category level, it would be necessary to test animate and inanimate subordinate categories that are not included in the training. One could, for example, train on bear, bison, elephant, hamster, hare, and test on lion, owl, tabby cat. If training on animate subordinate categories improves performance on the untrained animate subordinate categories, but not on the untrained inanimate subordinate categories, then we can conclude with confidence that training indeed improves performance at the superordinate category level. Without doing so, I am not sure it is valid to make such a claim. I am not trying to be difficult here, and I would like to be convinced. It should be easy to at least present some CNN results with such a manipulation?

Reviewer #3: I thank the authors for their engagement with my previous comments and for providing new analyses. While some concerns have been addressed, I still have a few important issues that need to be addressed:

1. Regarding category-specificity, I remain unconvinced by the authors' response. Given that the paper makes general claims about "category specificity" rather than just "animate vs. inanimate category specificity," I believe it's reasonable and within scope to demonstrate category specificity for at least a few additional categories in the ANN models. While collecting new human data would be outside the scope of the current work, testing other categories in the computational models should be feasible. If the authors disagree, I request a more substantial justification.

2. Regarding the layer freezing experiment conducted in response to my previous comment, the selection of layers appears somewhat arbitrary: layers 1-12 (13 layers) were frozen as "early" layers, while layers 5-19 (15 layers) were frozen as "deep" layers. These ranges, in my view, are too extensive to be meaningfully categorized as either "deep" or "early." Each experiment freezes approximately 70-75% of the network's layers. Could the authors clarify their rationale for determining which layers to freeze?

Additionally, when intermediate and deep layers are frozen (layers 5-19), we observe category specificity in the SSNR threshold of deep layers that mirrors the pattern seen when freezing layers 1-12, though with relatively smaller magnitude. The two panels in the third column of the new Figure 5 display very similar patterns in the deep layers. This suggests that the differential change in layers' SSNR cannot fully explain the category specificity, since the same differential pattern emerges in deep layers even when they're frozen. This requires further explanation.

Since this freezing experiment is the only reliable evidence for the layer-dependency of the category specificity, I believe it needs to be conducted exhaustively and clearly explained.

3. Regarding my question about individual vs. group-level analyses: I understand that specificities are measured in pre- vs. post-tests, but since these tests are conducted on each subject individually, it should be possible to measure the robustness effect within subjects, in addition to the group-level results currently presented. I'd appreciate clarification on why this individual-level analysis isn't included or feasible.

**Have the authors made all data and (if applicable) computational code underlying the findings in their manuscript fully available?**

Reviewer #1: Yes

Reviewer #2: Yes

Reviewer #3: Yes

PLOS authors have the option to publish the peer review history of their article (what does this mean? ). If published, this will include your full peer review and any attached files.

**Do you want your identity to be public for this peer review?** For information about this choice, including consent withdrawal, please see our Privacy Policy .

Reviewer #1: **Yes: ** Gregory Zelinsky

Reviewer #2: No

Reviewer #3: No

**Figure resubmission:**
---

## [Decision Letter · Decision Letter 2]

7 Sep 2025

PCOMPBIOL-D-24-01887R2

Category-specific perceptual learning of robust object recognition modelled using deep neural networks

PLOS Computational Biology

Dear Dr. Jang,

Thank you for submitting your manuscript to PLOS Computational Biology. After careful consideration, we feel that it has merit but does not fully meet PLOS Computational Biology's publication criteria as it currently stands. Therefore, we invite you to submit a revised version of the manuscript that addresses the points raised during the review process.

Specifically, note that R2 is making an important point: The results presented here primarily support the idea that learning in the model is specific to the basic object categories trained on, and does not show generalization to other categories either from the same or a different superordinate class (with one possible exception for training on animate1). Yet, the framing of the paper strongly implies that learning occurs on the level of the superordinate category. For example, the abstract says that humans "showed additional category-specific improvement". This would indicate that improvement occurs for objects within the trained category (even if the objects themselves were untrained), the same way that 'object-specific improvement' would be assumed to mean that performance improves even on untrained individual images of a trained object. Unfortunately, the human experiments do not allow us to determine whether human learning is actually occurring at the super or basic level (though previous uncited work suggests that inanimate training does not fully generalize to new inanimate objects: https://www.sciencedirect.com/science/article/pii/S0042698999001340). To fix this confusion, I do think it is reasonable to ask the authors to update language throughout the manuscript so as to make clear that this work does not speak to learning specificity on the super ordinate level, but is rather a test of basic object-level perceptual learning (bringing the discussion about Fig S2 into the Results rather than the Discussion may help with this, but I will leave that up to the authors). Separately, please also include the link to code in the manuscript itself.

Please submit your revised manuscript within 30 days Nov 07 2025 11:59PM. If you will need more time than this to complete your revisions, please reply to this message or contact the journal office at ploscompbiol@plos.org. Please include the following items when submitting your revised manuscript:

We look forward to receiving your revised manuscript.

Kind regards,

Grace W. Lindsay

Guest Editor

PLOS Computational Biology

Hugues Berry

Section Editor

PLOS Computational Biology

**Additional Editor Comments:**

Reviewer #1:

Reviewer #2:

Reviewer #3:

**Journal Requirements:**

1) Some material included in your submission may be copyrighted. According to PLOSu2019s copyright policy, authors who use figures or other material (e.g., graphics, clipart, maps) from another author or copyright holder must demonstrate or obtain permission to publish this material under the Creative Commons Attribution 4.0 International (CC BY 4.0) License used by PLOS journals. Please closely review the details of PLOSu2019s copyright requirements here: PLOS Licenses and Copyright. If you need to request permissions from a copyright holder, you may use PLOS's Copyright Content Permission form.

Potential Copyright Issues:

i) Please confirm (a) that you are the photographer of 1, or (b) provide written permission from the photographer to publish the photo(s) under our CC BY 4.0 license.

**Reviewers' comments:**

Reviewer's Responses to Questions

**Comments to the Authors:**

Reviewer #1: The additional analyses strengthened the manuscript. The researchers satisfyied all remaining concerns.

Reviewer #2: I thank the authors for their replies. Unfortunately, I am more confused than I was before.

(1) In the abstract, it is stated that “we sought to determine whether perceptual training with animate (or inanimate) object images in Gaussian noise would lead to category-specific or category-general improvements in robustness”. Here, it follows naturally that “category” refers to the animate vs. inanimate superordinate categories, and that a category-general improvement would mean crossing the animate vs. inanimate superordinate categories. If this is not the intended meaning, then the writing is very misleading and needs to be clarified throughout the manuscript.

Further down in lines 219-223, it says that “Observers trained on animate categories exhibited improved robustness when tested on animate categories (trend observed in 16/16 participants, t(15) = 8.23, p < 0.0001), but this improvement did not extend to inanimate categories (t(15) =0.81, p = 0.43), as was further supported by a significant statistical interaction effect (within-subjects ANOVA, F(1, 15) = 26.4, p = 0.0001).” Here again, categories refer to superordinate animate and inanimate categories. A direct read of this sentence conveys the meaning that training can transfer within the animate categories but not the inanimate categories. However, since untrained animate categories were not tested, this statement is misleading. The newly reported results indicate that such transfer is in fact very limited, absent in 3 out of the 4 cases.

It is worth noting that I am not the only one with such an interpretation. Review 3 appears to form the same interpretation.

(2) If the intended categories are the basic level categories, as the authors clarified in the reply letter, then we may have a bigger problem here, as whether training is category-specific or category-general would depend on what the test categories are. If the test categories are close to the trained categories (e.g., animate to animate), then we may see some improvement, indicating category-general effects, as shown by the newly reported results. However, if the test categories are far from the trained categories (e.g., animate to inanimate), then we see no improvement, indicating category-specific effects. Thus, whether the effect of training is category-specific or category-general seems somewhat of a moot question.

Reviewer #3: I thank the authors for their thorough response to my questions. The additional analyses completely address my concerns.

**Have the authors made all data and (if applicable) computational code underlying the findings in their manuscript fully available?**

Reviewer #1: Yes

Reviewer #2: **No: ** I did not see any statement about data sharing.

Reviewer #3: Yes

PLOS authors have the option to publish the peer review history of their article (what does this mean? ). If published, this will include your full peer review and any attached files.

**Do you want your identity to be public for this peer review?** For information about this choice, including consent withdrawal, please see our Privacy Policy .

Reviewer #1: **Yes: ** Gregory Zelinsky

Reviewer #2: No

Reviewer #3: No

**Figure resubmission:**
---

## [Editor Report · Decision Letter 3]

16 Sep 2025

Dear Professor Jang,

Thank you for thoughtfully incorporating the critiques of all reviewers. We are pleased to inform you that your manuscript 'Category-specific perceptual learning of robust object recognition modelled using deep neural networks' has been provisionally accepted for publication in PLOS Computational Biology.

Best regards,

Grace W. Lindsay

Guest Editor

PLOS Computational Biology

Hugues Berry

Section Editor

PLOS Computational Biology

---

## [Editor Report · Acceptance letter]

PCOMPBIOL-D-24-01887R3

Category-specific perceptual learning of robust object recognition modelled using deep neural networks

Dear Dr Jang,

I am pleased to inform you that your manuscript has been formally accepted for publication in PLOS Computational Biology. Your manuscript is now with our production department and you will be notified of the publication date in due course.

With kind regards,

Anita Estes
